# CAPR: Coherent Alignment of Constrained Reasoning Chains with Checklist-Driven Preference Refinement

## Abstract

Large language models (LLMs) with Chain-of-Thought (CoT) reasoning have shown remarkable capabilities in recent years, while domain adaptation through supervised fine-tuning (SFT) and reinforcement learning (RL) has become a common practice. However, these methods face significant challenges as the unconstrained CoT reasoning often leads to hallucinations, while RL techniques such as Direct Preference Optimization (DPO) suffer from alignment inefficiencies. In this work, we propose a unified framework to address these limitations by incorporating a domain-constrained reasoning paradigm and multi-dimensional preference alignment. Our approach introduces Domain-Constrained CoT Supervision, which integrates task-specific reasoning templates to enforce logical consistency and adaptability, along with Checklist-Driven Preference Refinement, which evaluates responses across orthogonal dimensions to provide precise signals for stable policy optimization. Extensive offline evaluations on large-scale industry datasets demonstrate the superior performance of our method in terms of factual accuracy. The rigorous online A/B tests confirm its ability to enhance conversation and selling strategy: +8.29% user Retention Rate, +2.19% Average Conversation Turns and +2.32% Order Rate.

## 1 Introduction

The rapid evolution of LLMs has unlocked unprecedented capabilities in general-purpose language understanding. Nevertheless, significant challenges remain in adapting these models to particular domains that require not only rigorous reasoning grounded in domain specific understanding but also precise alignment with complex human preferences. A widely recognized paradigm for extending LLMs to such domains involves SFT to instill domain-specific knowledge, followed by RL to refine alignment with human preferences Herold et al. (2025); Fang et al. (2025); Cheng et al. (2024). However, this two-stage approach faces significant limitations in both stages, which hinder its generalization and robustness. In the SFT stage, although effective at knowledge acquisition, current methods often struggle with the instability of the CoT process. Despite the introduction of techniques such as CoT distillation and process supervision to capture reasoning pathways, these approaches often face critical shortcomings due to the loose coupling between supervision and reasoning steps, resulting in issues like hallucinations and rigid, unnatural outputs Koksal & Alatan (2025); Gong et al. (2024); Gállego et al. (2025). Moreover, the lack of mechanisms to guide reasoning to domain-relevant strategies and logic further amplifies variability, reducing the applicability of trained LLMs in mission-critical tasks. At the RL alignment stage, efficiency bottlenecks and suboptimal convergence remain prominent. Specifically, as a widely adopted strategy, DPO faces significant constraints arising from the practical challenges of constructing accurate reward models. Moreover, self-sampling approaches often yield preference pairs with limited divergence, as both responses in a pair tend to share similar reasoning flaws. This phenomenon aligns with theoretical insights from preference learning, yet practical implementations fail to resolve the issue, leading to noisy gradient signals, inefficient policy optimization, and unstable training dynamics.

To address these challenges, we propose Coherent Alignment with Checklist-Driven Preference Refinement (CAPR), a novel two-stage framework designed to alleviate the above weaknesses. CAPR introduces domain-specific coherence at the reasoning stage and targets high-quality preference re-

finement during alignment, enabling a more robust and scalable approach for domain adaptation. In the SFT stage, we propose a Domain-Constrained CoT Supervision (DCCS) strategy to tightly align task-specific reasoning processes with human expert logical patterns. Specifically, we synthesize domain related CoT chains by decomposing human expert response strategy into structured reasoning steps, thereby effectively distilling human logical reasoning into the model. This strategy further restricts the sampling domain of each reasoning step, leading to increased stability, enhanced domain-specific expertise, and improved CoT performance. In the RL alignment stage, we introduce a Checklist-Driven Preference Refinement (CDPR) approach to systematically enhance preference optimization. CDPR leverages domain-specific checklists to evaluate generated responses across multiple dimensions, identify specific flaws, and construct corrective hints. These hints serve as conditioning factors for generating superior responses, yielding high-quality preference pairs (e.g., hint-corrected response vs. flawed response) with higher contrastiveness. This process significantly amplifies the informativeness and efficiency of DPO's supervision signals, directly addressing the inefficiency of standard self-evolution loops.

We rigorously validate the proposed CAPR framework through large scale experiments conducted in a challenging e-commerce customer service setting. Our approach demonstrates substantial improvements across both offline objective metrics (e.g., instruction adherence, correctness) and online subjective metrics (e.g., anthropomorphism, proactive problem resolution) compared to existing LLM-based baselines. Furthermore, upon deployment in a real-world e-commerce customer service environment, CAPR consistently outperforms competitors, achieving superior interaction quality and higher order conversion rate. The key contributions of our work are as follows:

- Domain-Constrained CoT Supervision: We propose a novel supervision strategy that integrates task-specific reasoning structures to enhance logical coherence and domain professionalism, resulting in models with more stable and interpretable reasoning processes.

- Checklist-Driven Preference Refinement: We introduce a preference refinement strategy to construct high-contrast, targeted preference pairs via checklist-driven good response generation, producing stronger gradient signals for efficient and stable preference optimization under a self-evolution paradigm.

- Comprehensive Evaluation and Real-World Impact: Through extensive experiments on both offline metrics and real-world deployment in commercial platforms, we demonstrate the efficacy and practical benefits of CAPR, achieving consistent business performance improvements such as increased order conversion rates.

## 2 RELATED WORKS

**Adapting Pretrained Reasoning Models.** The prevailing paradigm of adapting pre-trained LLMs through supervised fine-tuning on input-output pairs has been extensively studied Wei et al. (2021); Chung et al. (2024); Ouyang et al. (2022). However, critical challenges persist in transferring sophisticated reasoning capabilities from foundation models and ensuring strategic stability in open domain problem-solving Suzgun et al. (2022); Bubeck et al. (2023)—particularly given the established correlation between reasoning fidelity and downstream performance Lyu et al. (2023).

**Chain-of-Thought Paradigms.** Building upon the foundational work of Wei et al. (2022) that introduced CoT prompting, recent advances employ synthetic supervision through pseudo CoT distillation Magister et al. (2022); Ho et al. (2022) and task specific variants Zhou et al. (2022); Wang et al. (2022); Fu et al. (2022). While these methods enhance consistency, they remain constrained by their narrow focus on specialized domains and lack structured mechanisms to enforce domain aware reasoning, a crucial requirement for handling open-question scenarios where professional grounding directly impacts solution validity Nori et al. (2023). Our DCCS addresses these limitations through systematic integration of human expert logic patterns with synthesized domain constraints during fine-tuning, establishing structured professional reasoning as an explicit learning objective via contrastive trajectory optimization.

**Preference Alignment Techniques.** The evolution from Reinforcement Learning with Human Feedback (RLHF) Ouyang et al. (2022) to DPO Rafailov et al. (2023) has streamlined alignment through implicit reward modeling, driving widespread industrial adoption Touvron et al. (2023). Nevertheless, the effectiveness of such approaches hinges on the quality of contrastive pairs, a fun-

damental weakness when relying on naive self-sampling strategies that frequently generate indistinct examples with limited pedagogical value Alemohammad et al. (2024); Liu et al. (2023). Our CDPR introduces a paradigm shift through reasoning-aware data curation, where the model's intrinsic reasoning traces guide targeted preference pair generation. This methodology amplifies learning signals by strategically combining error diagnosis with logical path reinforcement, effectively addressing the signal dilution prevalent in conventional approaches.

## 3 METHOD

The primary objective of our method is to fine-tune a model $\pi_\theta(y|x)$ to adapt it to specific domain by instilling domain-specific knowledge and domain expert response strategy. We introduce a two-stage framework that first integrates human-logical reasoning patterns with synthesized domain-constrained reasoning chain, then refines the model's response by generating high contrastiveness preference pairs from checklist-driven conditional inference. The general setting of this problem consists of the input prompt $x$ (consists of instruction and input), a targeted response $y$ (human gold response) and the model $\pi$ with parameter $\theta$. The model inference process can be represented as conditional probability $\pi_\theta(y|x)$. Denote the CoT chain as $z$, this process can be further expressed by joint distribution $\pi_\theta(z, y|x) = \pi_\theta(z|x)\pi_\theta(y|x, z)$.

### 3.1 DOMAIN-CONSTRAINED COT SUPERVISION

The standard SFT approach maximizes the marginal log-likelihood $\log \pi_\theta(y|x)$ to fine-tune the model output with gold response. However, for the reasoning models, this objective lacks explicit constraints on the model's internal reasoning process, often leading to models that learn spurious correlations with hallucination. To mitigate this, we introduce a domain-constrained CoT chain $z$ and maximize the joint likelihood instead. The original marginal log-likelihood can be expressed as:

$$\log \pi_\theta(y|x) = \log \int \pi_\theta(y|x, z)\pi_\theta(z|x)dz. \tag{1}$$

This integral is intractable. Using variational inference, we can introduce an approximate posterior $q(z|x, y)$ and derive the Evidence Lower Bound (ELBO):

$$\log \pi_\theta(y|x) \geq \mathbb{E}_{z \sim q(z|x,y)}[\log \pi_\theta(y|x, z)] - \text{KL}(q(z|x,y)\|\pi_\theta(z|x)). \tag{2}$$

Our framework is shown in Figure. 1, we leverage outside teacher model to synthesize a high-quality reasoning chain $z^*$ for each pair $(x, y)$ based on the original gold response and corresponding instruction and input. This can be viewed as using a deterministic variational posterior $q(z|x, y) = \delta(z - z^*)$, where $\delta$ is the Dirac delta function. Substituting this into the ELBO yields:

$$\log \pi_\theta(y|x) \geq \log \pi_\theta(y|x, z^*) + \log \pi_\theta(z^*|x). \tag{3}$$

Thus, maximizing the joint likelihood $\log \pi_\theta(y, z^*|x)$ is equivalent to maximizing a tight lower bound on the true objective, $\log \pi_\theta(y|x)$. The teacher model generated $z^*$ acts as an amortized variational proposal, transforming the problem of learning a complex marginal distribution into learning two simpler conditional distributions. This property is further emphasized by the CoT chain consists of multiple predefined steps $z^* = \{z_1^*, z_2^*, ..., z_K^*\}$, where $K$ is the total steps of the CoT process. Therefore, the refined marginal log-likelihood can be expressed as:

$$\log \pi_\theta(y|x) = \log \sum_i \pi_\theta(y|x, z_i^*)\pi_\theta(z_i^*|x) \tag{4}$$

which decompose the original process into a finite and stable inference flow. The discrete structure $z^* = \{z_1^*, z_2^*, ..., z_K^*\}$ allows us to expand this bound into:

$$\log \pi_\theta(y|x) \geq \log \pi_\theta(y|x, z^*) + \sum_{i=1}^{K} \log \pi_\theta(z_i^*|z_{<i}^*, x) \tag{5}$$

This decomposition creates a tighter bound because the teacher-generated discrete steps $z_i^*$ provide a more constrained variational family, reducing the gap between $q(z|x, y) = \delta(z - z^*)$ and

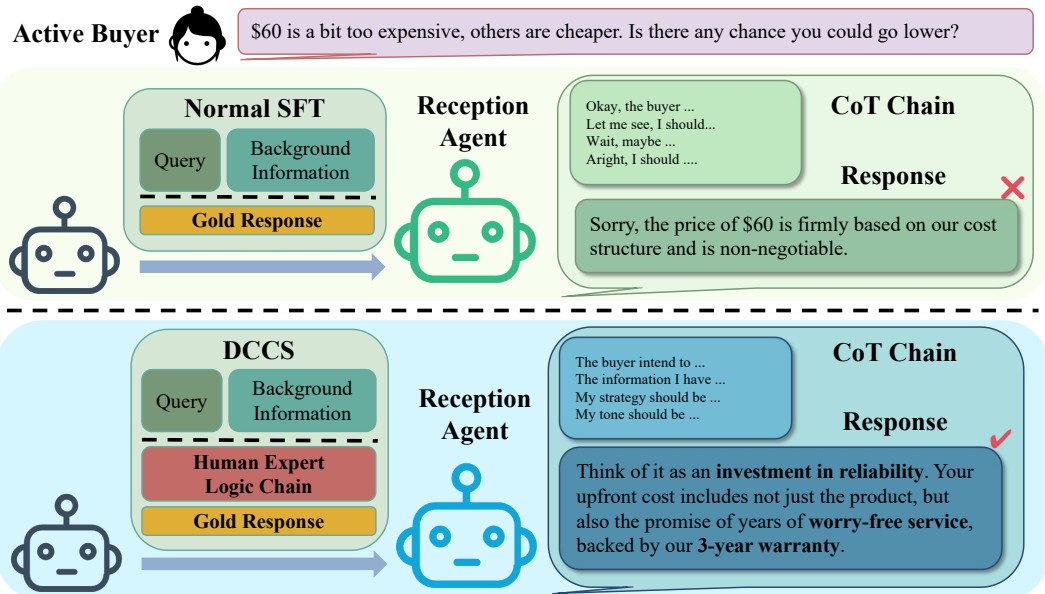

Figure 1: The illustration of our proposed Domain-Constrained CoT Supervision method. This method leverages domain-specific, human-expert logical chains to guide the reasoning process, thereby enhancing the stability of the language model and embedding expert-driven reasoning pathways into its generative capabilities.

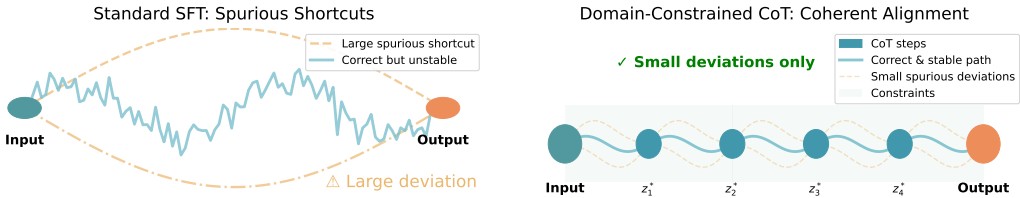

Figure 2: The reasoning patterns of a standard CoT model compared to our Domain-Constrained CoT model. By incorporating domain-constrained, step-wise supervision, our approach significantly reduces deviations in the reasoning process, resulting in improved stability and alignment with domain-specific logical structures.

the model distribution $\pi_\theta(z|x)$, as shown in Figure. 2. Rather than learning the entire chain distribution $\pi_\theta(z^*|x)$ at once—which may admit many spurious solutions—the factorized form constrains each step to be individually valid while maintaining their sequential dependencies. Each term $\log \pi_\theta(z_i^*|z_{<i}^*, x)$ provides a distinct optimization signal that guides the model toward the correct reasoning pattern.

Furthermore, the discrete formulation makes it easier for the model to approximate the delta distribution $\delta(z - z^*)$ by learning each step sequentially. By breaking down the complex reasoning distribution into $K$ simpler conditional distributions, each with lower complexity, we create multiple checkpoints where the model must align with the teacher-generated reasoning. This structured supervision prevents the model from taking shortcuts that would satisfy $\pi_\theta(y|x)$ without proper reasoning, thereby ensuring the learned distribution $\pi_\theta(y, z^*|x)$ better approximates the true joint distribution and results in a tighter bound on the marginal likelihood $\log \pi_\theta(y|x)$.

The final SFT loss function is a weighted sum:

$$\mathcal{L}_{\text{SFT}}(\theta) = - \sum_{(x,z^*,y)} \left[\log \pi_\theta(z^*|x) + \log \pi_\theta(y|x, z^*)\right], \tag{6}$$

The model inference process is therefore defined as:

$$P_\theta(Y|X) = P(Y|Z_K)(\prod_{i=2}^{K} P(Z_i|Z_{i-1}))P(Z_1|X) \tag{7}$$

This process has much higher stability and is much more learnable for the model.

## 3.2 CHECKLIST-DRIVEN PREFERENCE REFINEMENT

While DCCS introduces enhanced reasoning transparency and enforces domain-specific constraints, the model's ability to generate optimal responses can still degrade in complex scenarios characterized by high levels of uncertainty or ambiguity. Furthermore, aligning the model's outputs with human preferences during the RL stage remains a critical objective. To address this, we adopt DPO as an alignment method in domain adaptation for human interaction and conversational settings. This approach is particularly well-suited to scenarios where constructing a highly accurate reward model proves challenging. Specifically, DPO optimizes the model's policy by utilizing pairwise preference data $\mathcal{D} = \{(x, y^+, y^-)\}$, where $y^+$ corresponds to the preferred (chosen) response and $y^-$ represents the less optimal (rejected) candidate. The optimization process is guided by the following loss function:

$$\mathcal{L}_{\text{DPO}}(\theta) = -\mathbb{E}_{(x,y^+,y^-)\sim\mathcal{D}} \left[ \log \sigma \left( \beta \left( \log \frac{\pi_\theta(y^+|x)}{\pi_{\text{ref}}(y^+|x)} - \log \frac{\pi_\theta(y^-|x)}{\pi_{\text{ref}}(y^-|x)} \right) \right) \right], \tag{8}$$

where $\pi_{\text{ref}}$ denotes the reference model, commonly instantiated as the initial SFT model, and $\beta$ serves as the temperature parameter. The gradient of the loss function is proportional to $(1 - \sigma(\Delta))$, where $\Delta$ represents the preference difference between the response pair $\{y^+, y^-\}$. Importantly, when the positive $y^+$ and negative $y^-$ samples are highly similar ($\Delta \approx 0$), the resulting gradient signal becomes both weak and noisy Yang et al. (2025). This phenomenon can destabilize the training process, as $\pi_\theta(y|x)$ may simultaneously drift away from both the preferred and rejected examples.

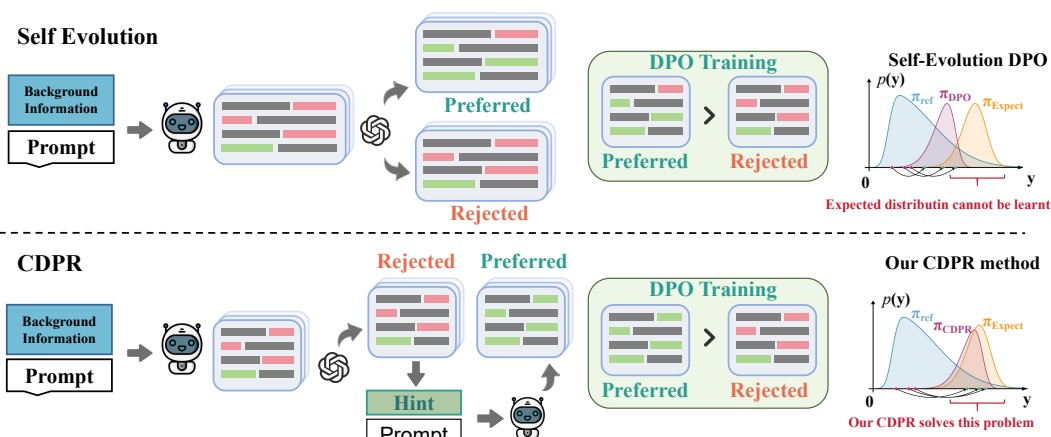

Figure 3: The general workflow of our proposed Checklist-Driven Preference Refinement method. Through CDPR operations, the generated responses are more effectively aligned with the expected distribution, facilitating more efficient and accurate learning within the DPO framework.

In this work, as shown in Figure. 3 we introduce a novel augmentation to DPO by incorporating checklist-guided preference optimization, a framework designed to systematically diagnose reasoning and response errors, as shown in. Our method generates high-contrast positive cases $y^+$ that are explicitly tailored to address the specific issues identified in negative cases $y^-$. By amplifying the contrast between preference pairs, this approach significantly improves both the stability and efficiency of DPO.

At the core of our approach is the explicit diagnosis of flawed reasoning and responses in the negative cases drived by the checklist. For a given input-response pair dataset, we begin by leveraging a predefined checklist for response $C = \{c_1, c_2, .., c_L\}$ consisting of $L$ criteria to filter the

bad case response pair $(x, y^-)$. Based on this pair, we can extract the associated reasoning chain $z^-$, which represents the sequence of intermediate reasoning steps derived as part of the DCCS framework. This reasoning chain is structured as $z^- = (z_1^-, z_2^-, \ldots, z_K^-)$, where $z_k^-$ denotes the $k$-th step in the chain. To evaluate $z^-$, we leverage another predefined checklist for reason chain $C^z = \{c_1^z, c_2^z, .., c_{L^z}\}$ consisting of $L^z$ criteria to filter the bad case response Each reasoning step $z_k^-$ is compared against the relevant checklist items to identify specific errors. For instance, flaws detected in the intent reasoning step $z_1^-$ may map to errors $c_1^z$, $c_2^z$, or a combination of both, while issues in deductive reasoning at $z_2^-$ might correspond to error $c_3^z$.

Each identified failure or flaw within the reasoning chain and response is systematically transformed into an actionable diagnostic textual suggestion, denoted as a hint $h_{i'}$, which specifies the weaknesses in the generated response with precision. The collection of these hints which derived from the checklist is represented as $h = \{h_{i'} | i' \in L^h\}$ where $L^h$ is the potential maximum hint number, encapsulating guidance for each reasoning aspect. These hints serve to instruct the model to refine its focus on the information or reasoning perspectives associated with the previously diagnosed flaws. By introducing this targeted intervention, the generative process is fundamentally augmented, auxiliary conditions (i.e. the hints) are incorporated alongside the original input, transforming $\{x\}$ into an enhanced input $\{x, h\}$, where $h$ provides explicit steering guidance. This augmentation enables the model to overcome spurious reasoning pathways and align more closely with robust, corrected responses.

The hint-augmented generation process ultimately yields a refined positive example $y^+$, designed to explicitly resolve the flawed elements present in the negative case $y^-$. Mathematically, this targeted intervention redefines the distribution over reasoning steps, shifting the prior $p(z|x)$ to a hint-conditioned posterior $p(z|x, h)$, thereby encouraging the generation of reasoning chains constrained in corrective guidance.

$$p(z|x, h) \propto p(h|z, x)p(z|x). \tag{9}$$

Here, $p(h|z, x)$ is a likelihood that assigns high probability to reasoning chains $z$ that are consistent with the hint $h$. This posterior then alters the distribution of final responses:

$$\pi(y^+|x, h) = \pi(y^+|x, z_K) \prod_{i=1}^{K} p(z_i|z_{<i}, x, h) \tag{10}$$

This intervention concentrates probability mass on high-quality responses that adhere to the corrective hint by guiding each discrete reasoning step $z_i$ toward the desired reasoning pattern. By constructing pairs $(y^+ \sim \pi(\cdot|x, h), y^-)$, we ensure that the positive sample is drawn from a distribution that is shifted towards higher-reward regions. This leads to a larger expected margin:

$$\mathbb{E}[\Delta_h] = \mathbb{E}[s_\theta(y^+) - s_\theta(y^-)] > \mathbb{E}[\Delta_{\text{self}}], \tag{11}$$

where $s_\theta(y) = \beta \log(\pi_\theta(y|x)/\pi_{\text{ref}}(y|x))$ is the reward score. A larger margin results in a stronger gradient signal $(1 - \sigma(\Delta))$, accelerating convergence. The Checklist-Hint DPO objective can then be formalized as:

$$\mathcal{L}_{\text{CDPR}} = -\mathbb{E}_{(x,h,y^+,y^-)} \log \sigma \left( \beta \cdot (r_\theta(y^+) - r_\theta(y^-)) \right), \tag{12}$$

Higher contrast $\Delta_h$ plays a pivotal role in mitigating the risk of vanishing gradients during preference updates, ensuring that the optimization process remains focused on meaningful corrections to the underlying reasoning.

A central strength of this framework is the stability it introduces into the training dynamics of DPO. By constraining $y^+$ to explicit reasoning corrections derived from diagnostic hints, rather than to random exploratory behaviors, the framework systematically reduces variance in positive case generation. This ensures that the alignment process maintains a stable range of rewards $r_\theta(y^+)$ and $r_\theta(y^-)$, bounded by a high-contrast difference, which effectively suppresses noisy updates throughout training. Empirically, this stabilization accelerates convergence and delivers faster optimization compared to naive implementations of DPO. Moreover, the design of the hint $h$ ensures it targets

specific errors within the reasoning chain, resulting in a positive response distribution $y^+$ that is less random and exhibits lower variance than outputs sampled from an unconstrained model, such as $\pi_{\text{SFT}}(y|x)$. This reduction in variance directly impacts the stability of gradient estimates, as lower variability in $s_\theta(y^+)$ propagates into the gradient updates, making the DPO optimization process substantially more efficient and stable. By leveraging the reasoning-guided diagnostic process, CDPR provides not only a stronger learning signal but also a more targeted corrective signal, directly addressing the model's weaknesses as detected through its reasoning trace.

In practice, we applied our method on the e-commercial scenario, with DCCS training with $K = 4$, which compose of intent identify, knowledge assessment, response strategy formulation, manner adjustment, to align with human logic chain. For the CDPR process, the potential maximum hint number $L_h$ is decided based on real business strategy, which compose of emphasizing factual information, proactive advancement, anthropomorphism and emotional engagement.

### 3.3 Implementation Details

In our e-commerce customer service setting, we adopt a four-step reasoning template aligned with expert decision logic: (1) intent identification, (2) knowledge assessment, (3) response strategy, and (4) manner adjustment. A strong teacher model (GPT-4o) rewrites responses into this structure via a deterministic JSON-constrained prompt (§A.1), requiring less than one expert-day for domain adaptation due to minimal human auditing.

Preference refinement is guided by a domain-specific checklist capturing key business qualities and the violations automatically produce corrective hints (§A.3), forming high-contrast preference pairs that enrich DPO signals. Ablations confirm that structured CoT and hint conditioning are both essential.

This design enables scalable deployment without additional annotation and generalizes effectively beyond the original domain, as demonstrated on GSM8K Cobbe et al. (2021) and ARC-Challenge Clark et al. (2018) benchmarks.

## 4 Experiment

### 4.1 Experiment Setup

Our foundational model is QwQ-32B Team (2025); Yang et al. (2024), which serves as the backbone for our proposed system deployed in a real-world e-commerce application. Specifically, the model is designed to process buyer questions based on product information, thereby guiding users purchase decisions. We conduct theoretical analyses and evaluate the model both offline and online to rigorously assess its performance. For offline evaluation, we utilize an e-commerce Q-A benchmark, comprising 10,000 question-answer pairs, to measure the model's response accuracy and consistency. For online evaluation, the model is integrated into the operational automatic customer service system, enabling direct interactions with buyers in live e-commerce settings. Through analysis of subsequent dialogues, we systematically examine the model's performance across various dimensions. Finally, an extensive A/B test is conducted to compare the commercial conversion rates of our system against GPT-4.1 Achiam et al. (2023) and LLama3 Grattafiori et al. (2024), the previously deployed baseline within the workflow.

To benchmark our full method, we evaluate against the following configurations: 1) Base: The original QwQ-32B model without any task-specific fine-tuning; 2) SFT: Standard Supervised Fine-Tuning on (input, response) pairs; 3) DCCS-SFT: Our proposed Domain-Constrained Chain-of-Thought Supervision approach applied during the fine-tuning stage; 4) DPO: The SFT model further refined via standard self-sampling Direct Preference Optimization; 5) CAPR: Our full proposed methodology, combining DCCS-SFT with Checklist-Driven Preference Refinement.

### 4.2 Evaluation Metrics

The offline evaluation framework encompasses multiple dimensions and tasks, including customization, logistics, and other relevant aspects of e-commerce operations. The proposed model processes the complete product dataset alongside the posed question to generate corresponding responses. To

assess the hallucination tendencies of the model, we employ GPT-4.1 to evaluta the correctness of the response.

In the context of online metrics, we establish a subjective evaluation benchmark aimed at assessing dialogue quality holistically at the session level. This involves criteria including conversation expertise, emotion engagement, proactive advancement, bottleneck-resolving and anthropomorphism. Each dialogue session is meticulously rated across predefined dimensions as good, normal, or bad, thereby enabling a nuanced assessment of conversational performance. We further compare our method with real responses generated by normal human sellers, top-performing human sellers, and an expert-curated set of labeled excellent responses extracted from conversations with top human sellers.

For real-world deployment, the model is integrated into a commercial platform to assist with buyer interactions. The primary performance indicators include the Order Rate on relation pair level, Average Conversation Turns (ACT), and Retention Rate. To gauge efficacy, we compute the relative change in these metrics compared to a baseline model (GPT-4.1), expressed as the percentage improvement or degradation relative to this established standard.

The theoretical analysis employs the t-SNE Maaten & Hinton (2008) method to reduce the dimensionality of the model output CoT and responses. We visualize the semantic distributions of the SFT model and the DCCS-SFT model outputs to the same 2-D plane under the same input conditions. We also calculate the statistical distribution of KL divergence between rejected response and chosen response for standard DPO pairs and checklist-driven DPO pairs.

### 4.3 EXPERIMENT RESULTS

Table 1: Comparison of Offline Question-Answering Accuracy (%) for all questions. Our completed method significantly out performs other baseline methods.

| Models | Prod-Attribute | Customization | Sample | Seller | Logistics | Payment |
|---|---|---|---|---|---|---|
| LLama3 | 36.7 | 51.9 | 32.5 | 26.4 | 23.1 | 7.2 |
| GPT4.1 | 57.6 | 65.7 | 46.0 | 43.5 | 46.9 | 13.5 |
| Base | 61.0 | 76.2 | 47.8 | 46.5 | 47.2 | 11.7 |
| SFT | 64.3 | 77.0 | 49.6 | 46.9 | 47.1 | 13.2 |
| DCCS-SFT | **72.3** | 78.3 | **53.4** | 47.5 | **44.9** | **15.5** |
| DPO | 72.1 | 77.4 | 50.7 | 47.2 | 41.5 | 14.7 |
| CAPR | 70.1 | **80.6** | 51.3 | **49.7** | 42.3 | 13.9 |

As shown in Table. 1, our model DCCS-SFT has shown most superior performance on all the tasks, which reveals the effectiveness of our training. In addtion, the DCCS-SFT model has shown better accuracy compare with the standard SFT model, which is majorly attributes to the relevant product information assessment step in the structured CoT. The CAPR model also outperforms the standard DPO model on most metrics, which proves the checklist-driven DPO training can provide higher comparison between good case answer and bad case answer. While the CAPR model encounters accuracy decline on some metrics after prefenrence alignment, this is mostly because we emphasize more on the proactive advancement and anthropomorphism on the RL stage. These results significantly demonstrated the additive benefit of each stage.

To better evaluate the model dialogue ability from the subjective perspective under real-world scenarios, we evaluated the dialogue quality in session level based on the conversation between our deployed model and customers. The evaluate is conducted from five aspects based on business negotiation requirements.

As shown in Table. 2, our completed method has shown good performance and conversation quality on most the aspects, which reveals the effectiveness of our training methods. The DCCS and CAPR consistently shown better performance compare with GPT-4.1, and even better than top human sellers, which proves both the effectiveness of the analyze process in the structured CoT and the checklist-driven preference refinement to align the model output with human preference.

To eventually assess the business improvements brought by our models, we deploy our trained model online and conducted the A/B Test experiments. As shown in Table. 3 our CAPR model yields sub-

Table 2: Comparison of online conversation subjective quality(/1). Our completed method shows outstanding dialogue ability on all the metrics.

| Models | Expertise | Emotion | Advancement | Bottleneck | Anthropomorphism |
|---|---|---|---|---|---|
| Human | 0.13 | 0.04 | 0.07 | 0.15 | 0.19 |
| Top-Human | 0.29 | 0.13 | 0.23 | 0.34 | 0.35 |
| H-Labeled | 0.63 | 0.46 | 0.67 | 0.47 | 0.68 |
| GPT4.1 | 0.27 | 0.09 | 0.24 | 0.25 | 0.17 |
| DCCS-SFT | 0.41 | 0.18 | 0.27 | **0.44** | 0.12 |
| CAPR | **0.43** | **0.21** | **0.32** | 0.39 | **0.21** |

Table 3: Comparison of Online Conversion Rates. Our A/B test compares our proposed model against GPT-4.1 used as the reception baseline. The reported percentages reflect the relative improvement or decline of our model compared to the GPT-4.1 baseline.

| Models | Order Rate | ACT | Retention Rate |
|---|---|---|---|
| DCCS-SFT | +1.03% | -1.05% | -0.18% |
| CAPR | **+2.32%** | **+2.19%** | **+8.29%** |

stantial improvements on all the business metrics, which is consistent with the previous evaluation. Especially for the retention rate, which is the crucially commercial signal, our method gains significant improvements. This property can be attributed to the advancement and anthropomorphism improvement of our method. This results show the real-world applicability of our models, and also prove the effectiveness of our design.

## 4.4 THEORETICAL ANALYSIS

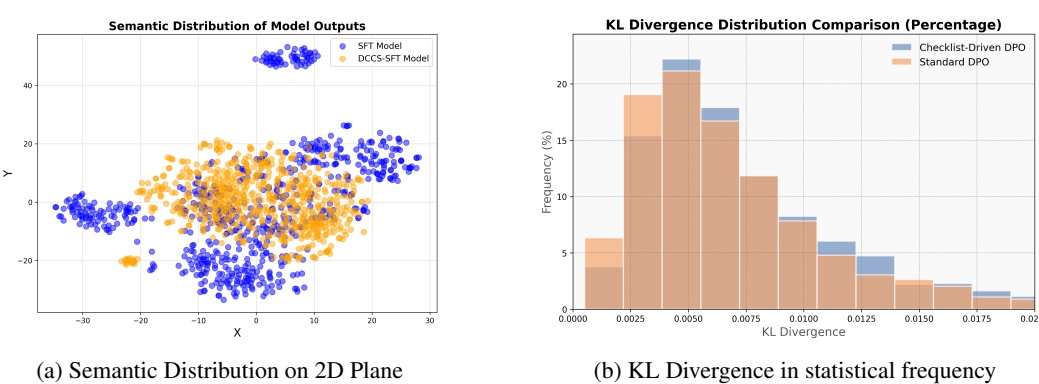

(a) Semantic Distribution on 2D Plane        (b) KL Divergence in statistical frequency

Figure 4: The visualization of the SFT model output distribution and the KL divergence of DPO preference pairs.

As shown in Figure 4a, we visualized the semantic distribution of the standard SFT model and our DCCS-SFT model outputs. As illustrated in Figure 1, the outputs of the DCCS-SFT model exhibit a significantly more compact distribution compared to the standard SFT model, validating our theoretical assumption that domain-constrained CoT training can make the model outputs and reasoning process more stable and domain-related. This property addresses the challenges faced by existing domain adaptation of reasoning models, namely instability and the lack of coherence induced by the CoT process.

For the DPO preference pairs, as shown in Figure 4b, we observe that checklist-driven DPO pairs consistently exhibit higher KL divergence compared to standard DPO pairs, resulting in improved preference optimization efficiency. This supports our assumption that, with checklist-driven hints, the model can provide better responses while operating within the self-evolution paradigm, further enhancing preference optimization.

## 4.5 EVALUATION ON GENERAL DATASETS

To validate the generalization of CAPR beyond e-commerce dialogue, we further evaluate on two standard reasoning benchmarks: (i) GSM8K Cobbe et al. (2021) for math word problems, and (ii) ARC-Challenge Clark et al. (2018) for grade-school science reasoning. We include strong CoT supervision (Alphamath Chen et al. (2024)) and recent preference optimization (KTO Ethayarajh et al. (2024)) methods for comparison.

| Dataset | Model | Base | SFT | Alphamath | KTO | Ours |
|---|---|---|---|---|---|---|
| GSM8K | Qwen3-0.6B | 0.42 | 0.46 | 0.48 | 0.49 | **0.52** |
| | Qwen3-4B | 0.85 | 0.90 | 0.91 | 0.92 | **0.94** |
| ARC-Challenge | Qwen3-0.6B | 0.31 | 0.35 | 0.39 | 0.37 | **0.43** |
| | Qwen3-4B | 0.50 | 0.56 | 0.59 | 0.58 | **0.62** |

Table 4: Accuracy (%) on GSM8K and ARC-Challenge. Results are averaged over 3 seeds; std $\leq$ 0.2%.

As shown in Table 6, CAPR consistently outperforms all baselines on both datasets, confirming that structured CoT supervision and checklist-driven refinement improve reasoning quality beyond our original domain.

## 4.6 SENSITIVITY ANALYZE

We vary the number of reasoning steps in DCCS ($K = 1$ to $5$) on Qwen3-0.6B. We also compare CAPR with and without hint-conditioned refinement.

| Dataset | Base | SFT | K=1 | K=2 | K=3 | K=4 | K=5 | CAPR | CAPR w/o Hint |
|---|---|---|---|---|---|---|---|---|---|
| GSM8K | 0.42 | 0.46 | 0.38 | 0.44 | 0.48 | 0.49 | 0.49 | **0.52** | 0.50 |
| ARC-C | 0.31 | 0.35 | 0.30 | 0.33 | 0.37 | 0.39 | 0.40 | **0.43** | 0.40 |

Table 5: Ablation results of CoT step $K$ and hint refinement on Qwen3-0.6B. Results are averaged over 3 seeds; std $\leq$ 0.2%.

Increasing $K$ improves DCCS reasoning supervision and consistently enhances performance, while checklist-driven hint refinement provides additional gains by strengthening preference pair contrast. These results confirm that DCCS and CDPR are complementary: the former enforces structured reasoning, and the latter further boosts alignment through targeted corrections.

## 5 CONCLUSION

In this work, we introduce a novel two-stage framework for adapting LLMs to specific domains, achieving improved stability and enhanced coherence while better aligning with domain-specific requirements. Building on the standard paradigm of SFT combined with RL, our approach incorporates targeted innovations at both stages. Specifically, we propose Domain-Constrained Chain-of-Thought Supervision to guide the SFT process, explicitly aligning the model's internal reasoning with the logic employed by human experts, thereby strengthening the reliability of intermediate reasoning. For the RL stage, we develop a Checklist-Driven Preference Refinement strategy, which optimizes preference modeling by increasing the contrastiveness and clarity of feedback signals during DPO training. Extensive experiments conducted not only on a challenging e-commerce dialogue task—evaluated through offline benchmarks and real online deployment—but also on public reasoning datasets such as GSM8K and ARC-Challenge, demonstrate that the proposed framework consistently improves both reasoning robustness and response quality. In particular, the online deployment yields substantial improvements in key business performance indicators, underscoring the practical impact and scalability of our approach. Looking ahead, we plan to extend this framework to a broader range of domains and explore its application to other alignment-critical tasks, further advancing the practical adaptability of LLMs.

# 6 REPRODUCIBILITY STATEMENT

The evaluation data in this study is derived from real commercial product and seller information collected from a large e-commerce platform. For offline evaluation, we utilize genuine product and seller metadata, paired with carefully designed synthetic test cases to ensure coverage of representative scenarios. Online evaluation involves deploying the trained model as a seller agent to interact directly with human buyers in a live commercial environment. The training dataset consists of real product and seller descriptions as well as high-quality conversations between buyers and top-rated human sellers, which were anonymized prior to use. Due to privacy and business constraints, the raw data cannot be released publicly; however, we will provide a processed and anonymized subset upon request, containing equivalent structure and statistics to facilitate reproduction of our experiments. We use publicly available pretrained language models as the backbone. All experiments were conducted on NVIDIA A100 GPUs using PyTorch 2.6, CUDA 12.6, and Python 3.11. Random seeds are fixed across runs. The complete source code for data preprocessing, training, evaluation, and model deployment scripts will be released on GitHub upon acceptance of the paper. We will also provide instructions for reproducing all results, including environment setup, model checkpoints, and evaluation protocols. The pretrained CAPR model will be publicly downloadable after internal verification and acceptance.

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

## 7 APPENDIX

During the preparation of this work, the author utilized ChatGPT-4 to assist with content refinement. After using this tool, the author reviewed and edited the content as necessary and takes full responsibility for the final published material.

### 7.1 A. COMPLETE IMPLEMENTATION SUPPLEMENT

#### 7.1.1 TEACHER MODEL PROMPT FOR DOMAIN-CONSTRAINED CoT COLLECTION

We use GPT-4o as the teacher model. The exact prompt is provided below:

Listing 1: Prompt template for teacher model

```
You are a top sales consultant on a B2B platform. Using the provided
[BackGroundKnowledge] and [AgentReturnHistory], analyze buyer intent,
formulate the most suitable response strategy, and decide how to
coordinate with other Agents to gather necessary information or directly
respond to the buyer.

Workflow:
1) Intent Identification
2) Knowledge Assessment
3) Response Strategy
4) Manner Adjustment

Output Format:
{
  "CoT": "1. ...; 2. ...; 3. ...; 4. ...;",
  "Reponse": "...",
}

Additional Instructions:
- Deterministic decoding only
- Ensure correctness and business policy compliance
```

### 7.1.2 TEMPLATE INSTANTIATION — EXAMPLE TRANSFORMATION

Below, we present both the original and the reconstructed responses.

Listing 2: Example of CoT construction

```
1  Raw sample:
2  We can deliver quickly. Want to place an order.
3  Domain Constrained CoT Example:
4  "CoT": "1. The buyer is asking about delivery time, indicating purchase
       intention;
5          2. We know current stock and lead time;
6          3. Provide exact delivery details and encourage ordering;
7          4. Use a polite and warm tone.",
8  "Reponse":  "Hello, this item is in stock and can be shipped within 3
       days. Let me know if you'd like to proceed, happy to assist anytime!"
```

### 7.1.3 CHECKLIST CONSTRUCTION AND HINT MAPPING

To ensure reproducibility in preference refinement, we provide the complete checklist designs and the mapping from checklist violations to corrective hints.

**CoT-Level Checklist (Reasoning Correctness)** We verify whether the structured Chain-of-Thought aligns with the domain reasoning logic. Each checklist item is a binary function $C(y) \in \{0, 1\}$.

$$C = \begin{cases} \text{IA-1: explicit\_intent\_correct,} \\ \text{IA-2: implicit\_intent\_inferred,} \\ \text{KA-1: product\_knowledge\_verified,} \\ \text{KA-2: missing\_info\_detected,} \\ \text{RS-1: strategy\_logically\_consistent,} \\ \text{RS-2: agent\_selection\_valid,} \\ \text{MA-1: manner\_adjustment\_justified} \end{cases}$$

Thus:

$$c_i(y) = \begin{cases} 1, & \text{if CoT step } i \text{ satisfies domain logic} \\ 0, & \text{otherwise} \end{cases}$$

**Response-Level Checklist (Execution Quality)**

We evaluate communication quality of the final answer. Each checklist item is similarly binary $C^z(y) \in \{0, 1\}$:

$$C^z = \begin{cases} \text{FC-1: factual\_correctness,} \\ \text{FC-2: factual\_safety,} \\ \text{PA-1: proactive\_advancement,} \\ \text{AP-1: anthropomorphism,} \\ \text{EE-1: emotional\_engagement,} \\ \text{UM-1: user\_centric\_messaging} \end{cases}$$

$$C_i^z(y) = \begin{cases} 1, & \text{if response satisfies evaluation requirement} \\ 0, & \text{otherwise} \end{cases}$$

**Automatic Hint Generation Based on Checklist Violations**

Given a model-generated flawed response $y^-$, we detect all violated items and each violated item is mapped to a short corrective hint. For example:

$$
\text{hint}(c) = \begin{cases}
\text{"First, explicitly state the buyer's primary need."}, & c = \text{IA-1}, \\
\text{"Ask about missing key product details before suggesting a path."}, & c = \text{KA-2}, \\
\text{"Encourage the user's next action, such as placing an order."}, & c = \text{PA-1}, \\
\text{"Avoid assumptions or placeholders; use only verified information."}, & c = \text{FC-2}.
\end{cases}
$$

## 7.2   B. ADDITIONAL EXPERIMENTAL ANALYSIS

### 7.2.1   STATISTICAL SIGNIFICANCE ANALYSIS

We repeat all evaluations with 3 random seeds and report both mean and standard deviation. As shown in Table 6, the standard deviation is consistently $\leq 0.2$ *absolute percentage points* across datasets and training settings, confirming that our improvements are statistically robust rather than random fluctuations.

| Dataset | Model | Base | SFT | Alphamath | KTO | CAPR |
|---|---|---|---|---|---|---|
| GSM8K | Qwen3-0.6B | 0.42 | 0.46 | 0.48 | 0.49 | **0.52** |
|  | Qwen3-4B | 0.85 | 0.90 | 0.91 | 0.92 | **0.94** |
| ARC-Challenge | Qwen3-0.6B | 0.31 | 0.35 | 0.39 | 0.37 | **0.43** |
|  | Qwen3-4B | 0.50 | 0.56 | 0.59 | 0.58 | **0.62** |

Table 6: Performance (%) on GSM8K and ARC-Challenge. Results are averaged over 3 seeds; std $\leq 0.2$ abs. percentage points.

### 7.2.2   BASELINE SETUP AND HYPERPARAMETERS

For fair comparison, we include two recent strong baselines:

**(1) Alphamath:** CoT distillation using step-level rationales. We adopt official training prompts and tune for 1 epoch on each dataset (batch size 64, learning rate $2e-5$).

**(2) KTO:** Preference training with prospect-based loss formulation. We apply self-sampled preference data following the original strategy.

All models use identical training data and tokenization. Evaluation prompts follow official protocol for GSM8K and ARC-Challenge. CAPR uses the same SFT stage as DCCS for fair comparison.

### 7.2.3   REWARD CORRELATION AMONG CHECKLIST DIMENSIONS

To validate that our two-level checklist introduces complementary training signals, we measure pairwise correlation between reward scores produced by checklist classifiers.

$$
\rho_{ij} = \frac{\text{cov}(R_i, R_j)}{\sigma(R_i) \cdot \sigma(R_j)}
$$

Across datasets, correlations remain below $0.18$, indicating the CoT-level and response-level feedback are not redundant. This supports the design choice of using structured reasoning (DCCS) and response refinement (CDPR) jointly.

### 7.2.4   ABLATION PROTOCOL

We vary the number of reasoning steps $K$ in DCCS: $K \in \{1, 2, 3, 4, 5\}$. We further remove hint conditioning from CDPR (*w/o Hint*) to isolate its contribution. All other configurations remain identical to Table 5. The results show: (i) larger $K$ improves reasoning stability by incorporating more domain logic; and (ii) hint-based refinement yields an additional $1\% - 3\%$ accuracy gain, demonstrating stronger contrastive preference supervision.

## 7.3 C. Deployment Reproducibility

This appendix provides full online evaluation configurations, runtime cost, manual effort estimation, and release plans.

### 7.3.1 A/B Testing Configuration and Statistical Significance

Our model was deployed in a commercial B2B dialog system, where a 14-day A/B test was conducted with equal traffic allocation (50% CAPR vs. 50% baseline GPT-4.1 tuned). The primary business KPI, user retention rate, demonstrates a statistically significant uplift with $p < 0.005$ under a three-sigma confidence level (99.7% CI). To ensure transparency and evaluation reliability, we further report subgroup statistics that validate both traffic balance and variance stability: $N_{\text{control}} = 60{,}926$ with $\sigma_{\text{control}} = 0.9470$, and $N_{\text{treatment}} = 60{,}641$ with $\sigma_{\text{treatment}} = 1.0109$. Although this subgroup metric is auxiliary and not used as the primary decision indicator, its directional consistency reinforces the robustness of our deployment evaluation.

### 7.3.2 Inference Latency and Serving Overhead

Checklist execution is rule-based and runs on CPU in $< 3$ ms per turn. Thus CDPR introduces negligible additional runtime cost and does not increase GPU memory usage, ensuring suitability for real-time commercial deployment.

### 7.3.3 Training Hyperparameters and Evaluation Protocol

We follow standard QwQ fine-tuning configurations, using a learning rate of $2 \times 10^{-5}$, batch size of 128, AdamW optimizer, and a maximum of 8K training steps. Evaluation strictly adheres to established benchmarks, including exact-match scoring for GSM8K and accuracy on the ARC-Challenge testdev split. All experimental results are averaged over three random seeds, with standard deviation consistently $\leq 0.2\%$, confirming robustness against random fluctuations. During online deployment, we additionally enable safety guardrails through rule-based reject sampling for hallucination filtering and automatic fallback to the baseline model when constraint violations are detected, ensuring fully reliable real-time serving.

### 7.3.4 Template Release and Reproducibility Materials

Upon acceptance, we will release:

- anonymized teacher prompting templates and CoT JSON examples
- full response-level and CoT-level checklist definitions
- offline evaluation scripts and data processing utilities
- training configurations for all models

This provides full reproducibility without revealing proprietary data.

