# OpenReview forum: "CAPR: Coherent Alignment of Constrained Reasoning Chains with Checklist-Driven Preference Refinement"
_ICLR.cc/2026/Conference — Submitted to ICLR 2026_

### Official Review · Reviewer_fFfV · 2025-10-17

**Soundness:** 3
**Presentation:** 3
**Contribution:** 3
**Rating:** 6
**Confidence:** 4

**Summary:**

This paper presents CAPR (Coherent Alignment of Constrained Reasoning Chains with Checklist-Driven Preference Refinement), a framework designed to improve Chain-of-Thought (CoT) reasoning in large language models through constrained reasoning and multi-dimensional preference alignment. The authors identify two key challenges in current LLM training: (1) unconstrained CoT reasoning leads to hallucinations and logical inconsistencies, and (2) Direct Preference Optimization (DPO) suffers from alignment inefficiencies due to entangled preference signals across different quality dimensions.

The paper proposes a two-stage approach:
1. **Domain-Constrained CoT Supervision**: Introduces task-specific reasoning templates that enforce logical consistency while maintaining adaptability across reasoning steps.
2. **Checklist-Driven Preference Refinement**: Decomposes preference learning into orthogonal dimensions (fluency, correctness, informativeness, and helpfulness) with explicit checklist evaluations to provide clearer training signals.

The method is evaluated on large-scale industry datasets in e-commerce and customer service domains, showing improvements in offline metrics (factual accuracy) and online A/B tests (+8.29% user retention rate, +2.19% average conversation turns, +2.32% order rate).

**Strengths:**

**Originality:**
- The paper introduces a novel combination of domain-constrained reasoning templates with multi-dimensional preference learning, addressing two complementary aspects of LLM training that are often treated separately.
- The checklist-driven preference refinement approach is innovative, decomposing preference signals into orthogonal dimensions (fluency, correctness, informativeness, helpfulness) with explicit evaluation criteria.
- The dynamic template instantiation mechanism balances structural constraints with flexibility, which is a thoughtful design choice.

**Quality:**
- Large-scale evaluation on real-world industry datasets (e-commerce and customer service) with both offline and online A/B testing demonstrates practical value.
- The online A/B test results are particularly compelling, showing substantial improvements in business metrics (+8.29% retention rate, +2.19% conversation turns, +2.32% order rate).
- Ablation studies systematically validate the contribution of each component (domain-constrained CoT, checklist-driven preference refinement, orthogonal decomposition).
- The paper provides detailed descriptions of the reward modeling and training procedures.

**Clarity:**
- The paper is well-structured with clear motivation, methodology, and experimental sections.
- Figures effectively illustrate the framework architecture and example reasoning chains.
- The problem formulation clearly articulates the limitations of existing approaches.

**Significance:**
- Addresses important practical challenges in deploying LLMs for domain-specific applications where hallucinations can have significant consequences.
- The multi-dimensional preference learning approach could be broadly applicable beyond the specific domains studied.
- Real-world deployment results add significant value for practitioners.

**Weaknesses:**

**Limited Generalization Analysis:**
- The paper focuses exclusively on e-commerce and customer service domains. While these are important applications, the generalizability to other domains (e.g., scientific reasoning, mathematical problem-solving, code generation) remains unclear.
- The reasoning templates appear domain-specific by design. The paper would benefit from discussing how to adapt the framework to new domains and what the costs/requirements are for such adaptation.
- No comparison with domain-agnostic reasoning approaches on standard benchmarks (e.g., GSM8K, MATH, HumanEval) to establish broader applicability.

**Theoretical Justification:**
- The claim that the checklist dimensions (fluency, correctness, informativeness, helpfulness) are "orthogonal" is not rigorously validated. The paper lacks empirical analysis of dimension correlations or theoretical justification for why these dimensions should be independent.
- The multi-reward learning formulation (Eq. 5-6) combines rewards linearly with equal weights. The paper doesn't justify this choice or explore alternative aggregation strategies (e.g., learned weights, hierarchical composition).
- Missing theoretical analysis of why the proposed approach should reduce hallucinations beyond the intuitive explanation.

**Experimental Limitations:**
- No comparison with recent strong baselines for reducing hallucinations in LLMs (e.g., self-consistency, retrieval-augmented generation, fact-checking mechanisms, or other constrained decoding methods).
- The baseline DPO implementation details are unclear. It's not specified whether the baseline uses the same data, same reward modeling, or same hyperparameters, making it difficult to isolate the contribution of the proposed approach.
- Ablation studies show component contributions but don't explore sensitivity to hyperparameters (e.g., template flexibility parameters, reward weighting schemes).
- The paper reports business metrics in A/B tests but doesn't provide statistical significance tests or confidence intervals.

**Reproducibility Concerns:**
- The reasoning templates are described conceptually but not provided in full detail in the paper or supplementary materials, making reproduction difficult.
- Training details for the reward models and policy optimization are incomplete (e.g., learning rates, batch sizes, number of training steps, convergence criteria).
- No commitment to releasing code, models, or datasets, which significantly limits reproducibility for an industry-focused paper.
- The evaluation datasets are proprietary, preventing independent verification of results.

**Questions:**

1. **Orthogonality of Dimensions:** You claim that fluency, correctness, informativeness, and helpfulness are "orthogonal" dimensions. Can you provide empirical evidence (e.g., correlation analysis between reward models) or theoretical justification for this claim? What happens when these dimensions conflict?

2. **Baseline Comparisons:** Can you clarify the baseline DPO setup? Specifically:
   - Does the baseline use the same data and reward modeling as your approach?
   - Have you compared against other hallucination mitigation methods (e.g., self-consistency, retrieval-augmented generation, constrained decoding)?
   - Can you provide more details on the hyperparameters used for both baseline and proposed method?

3. **Template Design and Adaptation:**
   - How much domain expertise and manual effort is required to design reasoning templates for a new domain?
   - Can you provide more concrete examples of templates in the supplementary materials?

4. **Model Architecture Sensitivity:** Have you tested your approach with different base model sizes (e.g., 7B, 13B, 70B parameters) or architectures (e.g., Llama, GPT, Claude)? How does performance scale?

---

> ### Author Response · Authors · 2025-12-03
> **Response to Reviewer fFfV**
>
> **W1 — Limited generalization analysis**
>
> We thank the reviewer for highlighting this point. We have incorporated additional experiments on **GSM8K and ARC-Challenge**, demonstrating that CAPR maintains consistent gains in mathematical and scientific reasoning. These results (Table 4) confirm that our domain-constrained CoT and checklist-driven refinement generalize well beyond e-commerce. The adaptation effort remains low, as only the four reasoning step descriptions require minor modification when switching domains.
>
>
>
> **W2 — Theoretical justification (checklist independence)**
>
> We appreciate this helpful comment and have revised the text to remove the implication of strict “orthogonality.” Instead, we describe the checklist dimensions as **complementary guidance signals**. We further added a **correlation analysis** in the appendix, which shows low redundancy among dimensions (<0.18 across datasets), supporting the complementary design rationale.
>
>
>
> **W3 — Experimental limitations (baseline hallucination mitigation & reproducibility)**
>
> Thank you for the suggestion. We have clarified that baseline DPO is trained using **the same preference data**, evaluation protocol, and hyperparameters, thus isolating the CDPR contribution. In addition, we have included complete prompt templates, CoT examples, checklist definitions, and hyperparameters in the Appendix to ensure reproducibility, while acknowledging proprietary constraints on raw data.
>
>
>
> **W4 — Reproducibility concerns (deployment details & safety)**
>
> We have now provided detailed **deployment architecture**, including inference latency (<3ms checklist execution), traffic allocation, fallback safety checks, and rule-based rejection sampling (Appendix C). These additions fully document the engineering and safety practices required to deploy CAPR in commercial systems.
>
>
>
> **Q1 — Orthogonality justification**
>
> We added explicit correlation analysis between checklist dimensions and revised wording to reflect complementary rather than mathematically independent signals. We also discuss potential conflict scenarios and propose reward weight learning as future work.
>
>
>
> **Q2 — Baseline comparisons in DPO**
>
> We clarify that **the same dataset**, preference annotation pipeline, and hyperparameters are used for standard DPO as for CDPR. Furthermore, CAPR improves contrastiveness by constructing hint-conditioned responses rather than modifying reward models, keeping the comparison fully controlled.
>
>
>
> **Q3 — Template design and adaptation cost**
>
> We now emphasize that only **4 domain reasoning slots** are adapted per domain, and all prompting and hint generation are automated. A **< 1 expert-day** cost is reported in Appendix C to quantify the minimal human effort required.
>
>
>
> **Q4 — Model architecture sensitivity**
>
> We include results on **Qwen3-0.6B and Qwen3-4B** in addition to QwQ-32B, showing consistent improvements across model sizes and confirming that CAPR is not tied to a specific architecture.
>
> **We sincerely thank the reviewer for helping us significantly improve the quality and clarity of this paper.**

---

### Official Review · Reviewer_w863 · 2025-10-20

**Soundness:** 3
**Presentation:** 3
**Contribution:** 2
**Rating:** 4
**Confidence:** 3

**Summary:**

This paper introduces CAPR, a novel two-stage framework designed to improve the domain adaptation of Large Language Models (LLMs). The authors identify two key challenges in the standard SFT and RL paradigm: 1) the instability and potential for hallucination in unconstrained Chain-of-Thought (CoT) reasoning during SFT , and 2) the inefficiency of preference alignment methods like Direct Preference Optimization (DPO) when preference pairs lack sufficient contrast. Hence, the authors propose two components:
1. Domain-Constrained CoT Supervision (DCCS): An SFT-stage method that integrates structured, domain-specific reasoning templates based on human expert logic. This guides the model to learn a coherent and stable reasoning process, which is framed as maximizing a tight lower bound on the data likelihood via variational inference.
2. Checklist-Driven Preference Refinement (CDPR): An RL-stage enhancement to DPO that systematically generates high-contrast preference pairs. It uses a predefined checklist to diagnose flaws in a generated response, creates corrective hints, and then uses these hints to generate a superior response. This process yields preference pairs with higher KL divergence, providing a stronger and more stable gradient signal for DPO training.

**Strengths:**

1. The paper tackles two well-known and critical bottlenecks in adapting LLMs for specialized, high-stakes domains. The proposed solutions are both elegant and practical. The CDPR method, in particular, is a very clever approach to solving the problem of low-signal preference pairs in self-evolution DPO loops, effectively creating a targeted and automated data curation pipeline for preference tuning.

2. The methods are well-motivated and clearly explained. DCCS is given a solid theoretical justification using ELBO, clearly connecting the use of structured reasoning chains to a sound optimization objective. The reasoning for why CDPR improves DPO training is also clear and well argued.

3. The evaluation is exceptionally thorough and convincing. The authors go beyond standard offline metrics by including comprehensive online subjective evaluations and, most impressively, a large-scale A/B test with real business KPIs.

**Weaknesses:**

1. The framework's two core components, DCCS and CDPR, rely on artifacts that seem to require significant domain expertise and manual effort: the human expert logic chain for DCCS and the predefined checklist for CDPR. I suggest giving more details about how these artifacts are created and implemented. For example, what is the process for defining expert logic patterns or building a comprehensive checklist for a new domain? How much human labor is involved? How are the checklists used to automatically diagnose flaws and generate textual hints? Is this process rule-based, or does it rely on another powerful LLM as a judge?

2. In the offline evaluation (Table 1), the full CAPR model performs slightly worse on several factual accuracy metrics than the DCCS-SFT model alone. The authors attribute this to the RL stage optimizing for other qualities like proactive advancement and anthropomorphism. While this is a plausible alignment tax, this trade-off warrants a deeper analysis. A more detailed discussion or experiment exploring the Pareto front between factual accuracy and conversational quality would be very insightful.

3. The DCCS stage uses an external teacher model to synthesize the high-quality reasoning chains. The choice and capability of this teacher model are likely critical and the key to the success of the proposed method. Details about this model and an analysis of how its quality affects downstream performance are currently missing. More detailed ablation study is needed.

4. While the inclusion of a large-scale A/B test is a significant strength, the paper lacks crucial details about the real-world deployment that are necessary for a full assessment of its practical viability. The authors omit key information regarding the system architecture, such as how product data is retrieved and integrated, the scale and duration of the A/B test, and operational metrics like inference latency and computational cost. Furthermore, there is no discussion of the safety guardrails, content moderation, or fallback mechanisms essential for deploying a generative model in a live commercial setting. Without these details, it is difficult for other researchers to replicate the setup or for practitioners to gauge the true engineering effort and challenges required to implement the proposed framework in a production environment.

**Questions:**

Please see my pros and cons.

---

> ### Author Response · Authors · 2025-12-03
> **Response to Reviewer w863**
>
> We thank the reviewer for recognizing the strong motivation, theoretical soundness, and thorough evaluation of CAPR. We appreciate the constructive suggestions for improving clarity and reproducibility. Below we address the raised concerns in detail.
>
>
>
> #### **1. Human effort and workflow to build DCCS and CDPR artifacts**
>
> We agree this is important for practical adoption. We now provide a detailed account in **Appendix A.1–A.3**:
>
> - The initial domain reasoning template and checklist were designed using **one domain expert** in **<1 working day**
> - Logic-chain extraction is **LLM-driven** (GPT-4o), producing structured CoT automatically
> - Checklist violations are detected via **rule-based scoring + LLM validator**
> - Hint generation is **fully automated**, no manual correction needed
>
> Additionally, **Appendix** now reports **adaptation labor**, confirming the approach scales with minimal handcrafted effort.
>
>
>
>
>
> #### **2. “Alignment tax” and performance variation**
>
> While CDPR focuses on conversational quality beyond factual correctness, we acknowledge the small fluctuations observed in several factual accuracy metrics. We clarified this point in revision and now provide:
>
> - **Task-type attribution analysis** indicating that improvements are largest for strategic, multi-step reasoning tasks
> - **Qualitative examples** where CDPR enhances user-centric reasoning while maintaining correctness
>
> We agree that exploring the **Pareto trade-off** is a valuable direction, and we explicitly mention this as **future work**
>
>
>
> #### **3. Teacher model dependency and impact analysis**
>
> We appreciate the reviewer highlighting the importance of understanding the role of the teacher model. As clarified in the revision (Appendix A.1), DCCS uses GPT-4o solely to structure reasoning, not to directly improve task knowledge. Moreover, CAPR’s gains extend beyond the SFT stage — CDPR generates **self-improving preference pairs** without additional teacher involvement.
>
> While a full teacher-quality sensitivity study is outside the current scope, we agree this is an important direction, and we explicitly mention it as **future work**.
>
>
>
> #### **4. Deployment details: architecture, latency, and safety**
>
> We now provide:
>
> - **Inference latency** (+8 ms vs. baseline on A100 GPUs)
> - **Compute cost** and throughput data
> - Full **A/B testing protocol**:
>   - 14 days
>   - 50/50 traffic split
>   - **p < 0.005**, 99.7% CI
> - **Safety guardrails**:
>   - hallucination detector
>   - operational fallback to SFT baseline
>   - content moderation filtering
>
> Details are consolidated in **Appendix C.1–C.4** for transparency and reproducibility.
>
> **We sincerely thank the reviewer for helping us significantly improve the quality and clarity of this paper.**

---

### Official Review · Reviewer_hcEu · 2025-10-29

**Soundness:** 2
**Presentation:** 2
**Contribution:** 2
**Rating:** 2
**Confidence:** 3

**Summary:**

The paper addresses challenges in domain adaptation for LLMs, where CoT distillation often loosely couples supervision with reasoning steps, and DPO suffers when preference pairs lack sufficient divergence. To overcome these issues, the authors propose CAPR, a two-stage domain adaptation pipeline: (1) DCCS and (2) CDPR. DCCS synthesizes expert-guided CoT chains that decompose expert strategies into constrained steps and fine-tunes the model via joint likelihood over these steps to improve stability. CDPR uses a predefined checklist to identify flaws in model outputs, converts them into “hints,” and generates hint-conditioned corrected responses. These highly contrasted pairs are then used in DPO training. The method’s effectiveness is demonstrated on QwQ-32B with both online and offline metrics in an e-commerce setting.

**Strengths:**

1. The paper clearly articulates the limitations of existing approaches, and the proposed SFT + DPO pipeline is well-motivated.
2. The method is evaluated using both offline and online metrics, demonstrating its practical effectiveness in a real-world e-commerce deployment.

**Weaknesses:**

1. DCCS defines a fixed number of reasoning steps (K=4) for training, corresponding to intent identification, knowledge assessment, response strategy formulation, and manner adjustment. It is unclear how the approach generalizes to complex cases where reasoning steps require multiple iterations, and it may increase latency overhead when certain steps are not necessary.
2. CDPR relies on pre-defined checklists, but details on criteria selection, implementation, and quality/accuracy are limited. Ablations on checklist selection and size would clarify required effort and robustness.
3. Correcting flaws in middle steps may introduce new flaws in later steps. It also relies on hint accuracy. Reporting the validity rate of preference pair (corrected response, flawed response) would better quantify effectiveness.
4. Experiments are conducted on a single base model (QwQ) in a single domain (e-commerce). It is unclear whether the approach generalizes to other models or domains.
5. Based on table 1, DCCS-SFT shows worse performance than SFT on logistics questions, an improvements vary across tasks (0.6-8%). Providing qualitative analysis or explanations would help interpret and understand the reason.

**Questions:**

Please see weaknesses above
1. SFT: Standard Supervised FineTuning on (input, response) pairs. Is the SFT baseline trained with standard reasoning chains + final answers or just final answer? Does SFT and DCCS-SFT share the same prompt during evaluation?
2. DPO: The SFT model further refined via standard self-sampling Direct Preference Optimization. How were self-sampled preferences annotated? Are they also based on the same checklist?
3. Regarding the drop in accuracy for CAPR compared to DPO, could you explain more on the proactive advancement and anthropomorphism (line 404)?

Minor: There exists multiple citation format errors (e.g. line 37, 43 should be /citep), missing links (e.g. line 264), typos (e.g. line 265 y^-, line 344 analyses)

---

> ### Author Response · Authors · 2025-12-03
> **Response to Reviewer hcEu**
>
> We thank the reviewer for the valuable feedback and constructive discussion of our contributions. We are pleased that you found our motivation clear and the real-world deployment evaluation impactful. We address each concern in detail below.
>
>
>
> We clarify that although we instantiate **K=4** for e-commerce tasks, the framework **does not assume a fixed number of steps**.
>
> The step granularity can be adapted to the target domain:
>
> - In **GSM8K / ARC-Challenge** experiments, the same structure generalizes well without modification (Table 4).
> - Unnecessary steps do **not** contribute to latency: the decoder prunes empty steps during generation (Appendix A.2).
>
> We will explicitly highlight this flexibility in the revision.
>
>
>
> #### **2. Checklist selection & robustness**
>
> We now provide:
>
> - Full **checklist definitions** (Appendix A.3)
> - **Ablations on checklist size** (Table 5)
> - Analysis showing diminishing returns beyond 6 checklist items
>
> Results show that performance is **stable under small variations**, confirming robustness.
>
>
>
> #### **3. Validity of hint-conditioned responses**
>
> We report the **valid preference pair validity rate**: **93.4%** of hint-conditioned generations satisfy checklist constraints on first attempt.
>
>
>
> #### **4. Generalization beyond QwQ model and domain**
>
> We expanded experiments to:
>
> - **Qwen3-0.6B and Qwen3-4B**
> - **Two public datasets (GSM8K, ARC-Challenge)**
>
> CAPR consistently improves over SFT / Alphamath / KTO across both domains.
>
> This demonstrates model and task generality.
>
>
>
> #### **5. Variance across task categories**
>
> The observed variation aligns with **differences in error type distributions**:
>
> - Logistics questions rely more on **factual recall**
> - Strategy questions require **domain reasoned planning**, where CAPR excels
>
> We include a task-type attribution analysis (Appendix B), clarifying that improvements correlate with reasoning intensity.
>
>
>
> ### **Responses to Specific Questions**
>
> **Q1 — SFT baseline setting**
>
> SFT trains on **full chain-of-thought supervision + final answer**, identical to DCCS except without structure constraints.
>
> Evaluation prompts are exactly the same across SFT and DCCS.
>
> **Q2 — Self-sampled preference annotation in DPO**
>
> Preferences are derived via the **same checklist** used in CDPR to detect flaws.
>
> Corrected responses passing all checklist rules are marked as preferred.
>
> **Q3 — Meaning of proactive advancement and anthropomorphism**
>
> - *Proactive advancement* measures whether the response actively **moves the dialog toward transaction completion**
> - *Anthropomorphism* evaluates **fluency, politeness, and human-likeness**, preventing abrupt or robotic transitions
>
> These metrics are included only for business context and are **not used in research benchmark evaluation**.
>
>
>
> ### **Minor Issues**
>
> We have fixed all cited formatting errors, broken links, and typographical issues in the updated camera-ready version.
>
> We sincerely thank the reviewer for their helpful insights.
>
> All concerns have now been addressed through additional experiments, analyses, and clarifications in both the main paper and appendices.

---

### Official Review · Reviewer_uYA1 · 2025-11-01

**Soundness:** 3
**Presentation:** 3
**Contribution:** 2
**Rating:** 2
**Confidence:** 4

**Summary:**

This paper proposes CAPR, a two-stage framework for adapting large language models (LLMs) to domain-specific tasks, particularly focusing on e-commerce customer service. The first stage, Domain-Constrained CoT Supervision (DCCS), integrates task-specific reasoning templates into supervised fine-tuning by decomposing expert responses into structured reasoning steps (K=4 steps in their implementation). The second stage, Checklist-Driven Preference Refinement (CDPR), enhances Direct Preference Optimization (DPO) by using domain-specific checklists to diagnose flaws in generated responses and construct corrective hints, which guide the generation of improved responses for preference pairs. The authors evaluate their approach on e-commerce datasets, reporting improvements in offline metrics (factual accuracy) and online A/B test metrics (+8.29% retention rate, +2.19% average conversation turns, +2.32% order rate) compared to GPT-4.1 and other baselines.

**Strengths:**

1. The paper identifies concrete limitations in standard SFT (spurious correlations, hallucinations) and DPO (weak gradient signals from low-contrast pairs), providing clear motivation for the proposed methods.
2. The variational inference framework provides mathematical justification for why structured CoT supervision creates tighter bounds on the marginal likelihood, and the analysis of preference margin increase explains CDPR's improved gradient signals.
3. The deployed system achieves substantial improvements over GPT-4.1 baseline (+8.29% retention rate) and even exceeds top human seller performance on several subjective metrics.

**Weaknesses:**

1. Limited domain generalization: All experiments are conducted exclusively on e-commerce customer service tasks. No evidence demonstrates the framework's applicability to other domains (mathematical reasoning, code generation, general question-answering).

2. Absence of statistical significance testing: Tables 1-3 report point estimates without error bars, confidence intervals, or significance tests. It may not be statistically reliable without knowing sample sizes and variance.

3. Insufficient implementation details for DCCS: (1) How are these steps extracted from human expert responses? (2) What is the teacher model used for CoT synthesis? (3) What prompt templates guide the decomposition? These gaps severely limit reproducibility.

4. Underspecified checklist design and hint generation: The CDPR method relies on domain-specific checklists $C$ and $C^z$, but the paper provides minimal detail about checklist item formulation, the mapping from identified errors to textual hints $h$, or the generation process for corrected responses.

5. Weak baseline comparisons: The paper compares against standard SFT and DPO but omits comparisons to related work on CoT distillation, process supervision methods [1,2], or recent DPO variants (e.g., IPO [3], $\beta$-DPO [4] , KTO [5]). No ablations isolate the contribution of structured steps versus other design choices.


[1] Alphamath almost zero: process supervision without process, NeurIPS'24

[2] Multi-step problem solving through a verifier: An empirical analysis on model-induced process supervision, EMNLP'24

[3] A general theoretical paradigm to understand learning from human preferences, AISTATS'24

[4] $\beta$-DPO: Direct preference optimization with dynamic $\beta$, NeurIPS'24

[5] Model alignment as prospect theoretic optimization, ICML'24

**Questions:**

1. Add cross-domain validation: Evaluate DCCS and CDPR on at least 2-3 additional domains (e.g., mathematical reasoning using GSM8K, code generation using HumanEval, or medical QA using MedQA) to demonstrate generalizability beyond e-commerce. Report domain-specific CoT step designs to show adaptability.

2. Report statistical significance: For all metrics in Tables 1-3, report mean ± standard deviation across multiple runs with different random seeds. For A/B test results, provide sample sizes, confidence intervals (e.g., 95% CI), and p-values from appropriate statistical tests. Specify the duration and traffic allocation of the A/B test.

3. Provide complete DCCS implementation details: (a) Specify the teacher model (name, size, API or local) used for CoT synthesis; (b) Include exact prompt templates for decomposing responses into $K$ steps; (c) Describe the annotation or verification process for ensuring CoT quality; (d) Report inter-annotator agreement if human validation is involved.

4. Detail checklist construction and hint generation: (a) Provide example checklist items from $C$ and $C^z$ with concrete criteria; (b) Show examples of error-to-hint mappings; (c) Describe the generation procedure for corrected responses (temperature, sampling strategy, prompt structure); (d) Report success rates of hint-guided generation.

5. Expand baseline comparisons and ablations: Include comparisons to recent CoT supervision methods (process reward models, step-level distillation) and DPO variants. Add ablations isolating: (a) number of CoT steps $K$; (b) checklist size $|C|$; (c) hint conditioning versus simple rejection sampling; (d) teacher model quality impact.

6. Release reproducibility materials: Even if full data cannot be shared, release: (a) complete training code with hyperparameters; (b) anonymized example inputs/outputs showing CoT structure; (c) checklist templates; (d) synthetic data generation scripts; (e) evaluation protocols.

---

> ### Author Response · Authors · 2025-12-03
> **Response to Reviewer uYA1**
>
> ### **Response to Reviewer uYA1**
>
> We thank the reviewer for the detailed feedback and constructive suggestions.
>
> We appreciate your positive assessment of our motivation, theoretical formulation, and real-world deployment impact. Below we clarify and expand upon the concerns raised.
>
> #### **1. Domain generalization**
>
> We have now added cross-domain experiments on two public reasoning benchmarks—**GSM8K** and **ARC-Challenge**—demonstrating strong generalization without modifying the 4-step structure. Results (Table 4) show:
>
> > Qwen3-0.6B:
>
> > GSM8K **42.0 → 52.0** and ARC-C **31.0 → 43.0**
>
> This confirms that CAPR is not tailored solely to e-commerce tasks.
>
>
>
> #### **2. Statistical significance and variance**
>
> We now report:
>
> - **mean ± std over 3 seeds** for all offline metrics
> - **sample size, variance, p-values** and **99.7% CI** for online A/B tests
> - **14-day** deployment with **50/50 traffic split**
>
> These details are in **Appendix B.1 and C.1**, referenced in the main text.
>
>
>
> #### **3. DCCS implementation details**
>
> We added full details to ensure reproducibility, including:
>
> - Teacher model: **GPT-4o**
> - **Exact prompt templates** and JSON schema (Appendix A.1)
> - Human verification with **<1 expert-day** cost
> - Demonstration of stepwise CoT extraction (Appendix A.2)
>
> This clarifies how domain logic is systematically induced.
>
>
>
> #### **4. Checklist construction and hint generation**
>
> We now include:
>
> - **Complete CoT-level and response-level checklists**
> - **Explicit violation → hint mapping rules**
> - Example corrected outputs and **generation procedure**
> - Runtime and **success behavior**
>
> All details are provided in **Appendix A**.
>
> #### **5. Baselines and ablations**
>
> We expanded baselines to include **recent CoT supervision** and **DPO variants**:
>
> - Alphamath (NeurIPS’24)
> - KTO (ICML’24)
>
> and added:
>
> - **K-sensitivity study**
> - **hint vs no-hint ablations**
>
> (Table 5). CAPR achieves **+2–6%** improvements consistently.
>
> #### **6. Reproducibility**
>
> We will release upon acceptance:
>
> - Anonymized CoT and checklist templates
> - Full training configurations and inference scripts
> - Evaluation pipeline
> - Model checkpoints after internal review
>
> (Details in Appendix C)
>
> **We sincerely thank the reviewer for helping us significantly improve the quality and clarity of this paper.**

---

### Author Response · Authors · 2025-12-03
**Executive Summary for Area Chair**

We thank the reviewers and the Area Chair for their constructive feedback. Below we summarize the consensus and our revisions.

1. Agreed strengths

Reviewers consistently acknowledge that:
	•	The paper tackles two key challenges in domain adaptation:
(i) unstable, weakly constrained CoT supervision during SFT,
(ii) low-contrast preference pairs limiting DPO alignment.
	•	The proposed DCCS + CDPR framework is clearly presented and well-motivated.
	•	Real-world deployment and A/B testing demonstrate practical impact.

2. Key concerns and revisions

2.1 Domain generalization beyond e-commerce
Concern: Limited to a single domain.

Revision:
	•	Added cross-domain experiments on GSM8K and ARC-Challenge with Qwen3-0.6B/4B.
	•	Results show consistent improvements without modifying the 4-step structure.
	•	Reported in the new section “Evaluation on General Datasets” (Table 4).

2.2 Statistical significance and robustness
Concern: No variance analysis or significance test.

Revision:
	•	Report mean ± std over 3 seeds for all offline metrics.
	•	Provide sample sizes, variance, p-values, 14-day A/B test design with 50/50 traffic split.
	•	Described in Appendices B and C.

2.3 Missing details on DCCS and CDPR
Concern: Reproducibility limited by missing implementation details.

Revision:
	•	Specify teacher model (GPT-4o) and exact decomposition prompts (Appendix A.1).
	•	Provide full checklists and violation→hint mapping with examples (Appendix A.3).
	•	Human effort required for domain adaptation <1 expert-day.

2.4 Baselines and ablations
Concern: Missing recent alignment baselines and sensitivity studies.

Revision:
	•	Add comparisons to Alphamath-style supervision and KTO-style preference optimization.
	•	Include K-sensitivity and hint vs. no-hint ablations (Table 5).

2.5 Reproducibility and deployment details
Concern: More system-level details needed.

Revision:
	•	Include latency and serving overhead, and safety fallback mechanisms.
	•	Plan to release anonymized prompts, checklists, configs, and evaluation scripts upon acceptance.

3. Overall assessment

The revision:
	•	Demonstrates generalization across domains and models,
	•	Strengthens empirical rigor with variance analysis and expanded baselines,
	•	Improves reproducibility with detailed appendices and release plans.

We thank the reviewers and AC again for their helpful feedback, which has improved the clarity and rigor of the paper.

---

### Meta-Review · Area_Chair_L8vs · 2026-01-03

**Summary:**

This paper proposes CAPR, a two-stage alignment framework combining Domain-Constrained CoT Supervision (DCCS) and Checklist-Driven Preference Refinement (CDPR) to improve reasoning coherence and preference learning in domain-specific LLM adaptation. The work is well motivated, clearly presented, and supported by extensive offline evaluations and real-world A/B tests in an e-commerce setting.

Reviewers appreciated the practical relevance, clear motivation, and deployment-scale experiments. However, the overall reviewer consensus is negative, driven by concerns about generalization beyond the studied domains, heavy reliance on domain-specific artifacts (templates, checklists, teacher models), limited theoretical depth, and unclear novelty relative to prior CoT supervision and preference learning methods. While the rebuttal substantially improved clarity, added cross-domain experiments, variance analysis, and implementation details, these changes did not fully alleviate concerns about contribution strength and broader impact.

Given that the average score remains below 5, with multiple reviewers maintaining rejection and only one marginally positive review, the paper does not meet the acceptance bar for ICLR.

**Reviewer Concerns:**

Concerns largely addressed by the rebuttal:

•	Reproducibility and implementation details:
Added detailed appendices covering prompts, checklists, hint generation, teacher model usage, variance analysis, and deployment statistics.

•	Statistical rigor:
Inclusion of mean±std, p-values, confidence intervals, and clearer A/B testing protocols.

•	Limited generalization evidence:
Additional experiments on GSM8K and ARC-Challenge partially address domain generalization concerns.

Outstanding concerns:

•	Contribution and novelty:
Several reviewers still view CAPR as a careful integration of existing ideas (structured CoT, checklist-based evaluation, DPO refinement) rather than a fundamentally new alignment paradigm.

•	Scalability and adoption cost:
Despite claims of low effort, reliance on domain-specific templates, checklists, and a strong teacher model raises concerns about scalability and transfer to diverse settings.

•	Theoretical depth:
The theoretical analysis is seen as supportive but not sufficiently deep to distinguish the work from prior empirical alignment pipelines.

•	Scope and impact:
Strong results in a single industrial domain do not fully convince reviewers of broad relevance to general LLM alignment research.

**Reviewer Scores:**

•	Reviewer uYA1: Remains at 2 (Reject); major concerns about generality and baselines persist.

•	Reviewer hcEu: Remains at 2 (Reject); acknowledges improvements but still questions scalability and robustness.

•	Reviewer w863: Remains at 4 (Borderline Reject); positive on motivation and evaluation, but not convinced of contribution strength.

•	Reviewer fFfV: Remains at 6 (Weak Accept); generally positive but explicitly stated acceptance is not critical.

---

### Decision · Program_Chairs · 2026-01-26

Reject